# Study of the Mechanical Properties and Thermal Control Performance of Plasma-Sprayed Alumina Coating on Aluminum Alloy Surface

Gengchao He [1,2], Weiling Guo [2], Dongyu He [2], Jiaqiang Zhang [3], Zhiguo Xing [2], Zhenlin Lv [1], Lei Jia [1,*] and Yanfei Huang [2,*]

1   School of Materials Science and Engineering, Xi'an University of Technology, Xi'an 710048, China
2   National Key Lab for Remanufacturing, Army Academy of Armored Forces, Beijing 100072, China
3   Beijing Spacecrafts, China Academy of Space Technology, Beijing 100094, China
*   Correspondence: xautjialei@163.com (L.J.); huangyanfei123@126.com (Y.H.)

**Abstract:** Thermal control coating is an important means of ensuring that a spacecraft remains operational at high temperatures. Due to limitations regarding preparation technology and material properties, the mechanical properties of the conventional thermal control coatings still need to be improved. To solve this problem, nanostructured alumina coatings (NCs) and conventional alumina coatings (CCs) were prepared using plasma-spraying technology. The microscopic morphology, phase structure, hardness, and thermal control properties (solar absorptance ($\alpha_s$) and emissivity ($\varepsilon$)) of the nanostructured alumina coatings were investigated and compared with those of conventional alumina coatings. The results show that the NC has a higher hardness value (1168.8 HV) and that its reflectivity exceeds 75% in the wavelength range of 446–1586 nm, while a high degree of emissivity of 0.863–0.87 is still maintained at 300–393 K. Furthermore, the results show that these highly reflective properties are related to the phase composition and internal micromorphology of the NC, whereby the solar absorption of the coating is reduced due to the increase in the alpha phase content (21.4%), the high porosity (5.21%) and the nanoparticles favoring the internal scattering. All these properties can improve the performance of this CC coating with low solar absorptance ($\alpha_s$) and high emissivity ($\varepsilon$).

**Keywords:** plasma spraying; alumina; nanostructured; thermal control property





## 1. Introduction

High-strength aluminum alloys (2A12) are widely used as structural materials in spacecrafts to reduce their structural weight [1,2]. However, the equilibrium temperature of a spacecraft's surface is high under solar radiation due to the low emissivity of these alloys. Coating these surfaces with a layer of low-absorption, high-emission thermal control coating can reduce the absorption of solar radiation, increase surface thermal radiation, and reduce the thermal equilibrium temperature of the spacecraft, thus ensuring that the spacecraft and its various instruments and equipment can maintain a normal operating temperature range [3–5].

At present, the common thermal control coatings (TCCs) for aluminum surfaces are mainly divided into the white paint type [6–9], the second surface mirror type [10–13], and the electrochemical ceramic type [14–16]. The white paint type is mainly composed of white powders (e.g., ZnO and $TiO_2$) and binders such as epoxies, acrylics, silicones, and silicates, which are deposited on the substrate surface by coating or air-spray technology and cured to obtain a thermal control coating [8,9]. White paint coating offers the advantages of a simple preparation process, high efficiency, and easy repair. However, the degradation of the binder in an ultra-vacuum and alternating hot and cold environments leads to the hardening of the coating, which reduces the bonding properties of the coating to the substrate. The secondary surface mirror-type thermal control coatings, also known as

optical solar reflectors (OSR), are usually made by depositing $SiO_2$ thin films on metal substrates by vacuum vapor deposition [11,12]. OSR coatings have both high IR emissivity and low solar absorption and have shown excellent stability in practical applications. However, they are expensive and cannot be deposited on the surface of large, complex structural parts. In addition, secondary surface mirror-type coatings have poor wear resistance and, therefore, are prone to wear and even peeling in the harsh environment of space, resulting in degraded coating performance [13]. Ceramic thermal control coating is an aluminum oxide coating formed on the surface of an aluminum alloy by anodic oxidation or plasma electrolytic oxidation (PEO) [15,16]. In recent years, researchers have achieved the modulation of coatings' microstructure and composition by regulating the electrolyte composition and optimizing the reaction's electrical parameters, thus improving coatings' properties. However, this coating has poor high-temperature resistance and still needs to be improved in terms of its solar absorbance performance. In summary, due to limitations in terms of preparation technology and material properties, the mechanical properties of thermal control coatings still need to be improved.

Compared with the latter materials and processes, plasma-spraying technology has a high deposition rate and can deposit a variety of materials, from metals to ceramics, on substrates of any geometry and of different sizes and offers a strong binding force with the substrate [17–19]. As is commonly known, plasma-sprayed alumina coatings are widely used in a variety of fields due to their excellent physical and chemical properties [20–23]. Recently, studies have shown that coatings prepared by spray-granulated nano-alumina powder via atomization exhibit better mechanical properties than conventional coatings due to their unique microstructure [24,25]. As shown in Figure 1a, during the spraying process, the melting state of the granulated $Al_2O_3$ nanoparticles can be divided into three stages, namely, completely melted, partially melted, and unmelted, and these stages depend on their sizes and trajectory in the plasma jet [26,27]. Accordingly, Figure 1b shows that some of the original nanoparticles are retained in the microstructure of the coating, and these particles can effectively impede dislocation movement and improve the hardness and creep resistance of the coating [28]. However, detailed studies regarding the influence of microstructural features on the thermal control properties of nano-alumina coatings are limited in the literature. In general, the thermal control properties (solar absorptance ($\alpha s$) and infrared emissivity ($\varepsilon$)) of coatings are determined by their microstructure and composition [29,30]. J. Marthe et al. [29] optimized the scattering and reflective properties of porous alumina by controlling the spray parameters in order to optimize porosity. The absorption rate of the coating was less than 0.1 within 300–800 nm. In addition, Denis Toru et al. [30] found a significant increase in the reflectivity of an alumina plasma-sprayed coating caused by the conversion of $\gamma$-$Al_2O_3$ to $\alpha$-$Al_2O_3$ after an annealing treatment.

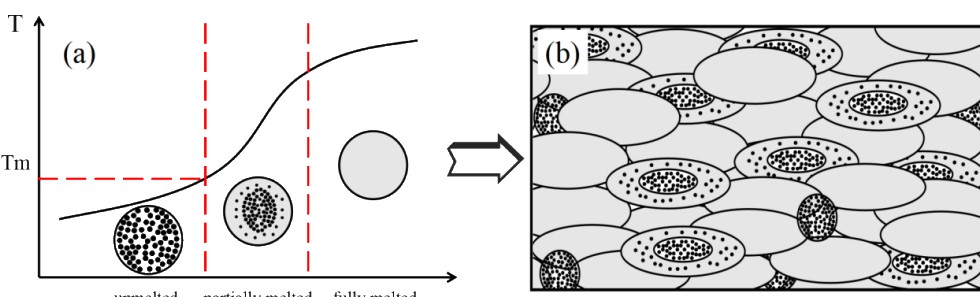

**Figure 1.** Schematic of formation process of plasma-sprayed nano $Al_2O_3$ coating. (**a**) Melting state of particles; (**b**) nano-sized $Al_2O_3$ particle-based coating models [25].

Therefore, in this study, we used an atmospheric plasma-spraying method to prepare conventional alumina coatings and nanostructured alumina coatings in order to character-ize the microstructures, thermal control properties, and mechanical properties of different

alumina coatings and to relate their microstructures and composition with their thermal control and mechanical properties.

## 2. Materials and Methods

### 2.1. Powder Characterization

As shown in Figure 2, scanning electron microscopy (SEM) (Supra55, Zeisi, Germany) and Mastersizer 2000 laser particle size analyzer were used to characterize the morphology and size statistics of two $Al_2O_3$ powders (conventional micron powder and nano-powder after granulation) selected for spraying experiment. We determined that the particle size distributions of conventional powder and granulated nano-alumina powder obtained via atomization are in the range of 20–100 μm and 20–80 μm, respectively, with an average particle size of (Dv50) 52.1 μm and 40.8 μm, respectively. The morphologies of the powders are angular and spherical, respectively.

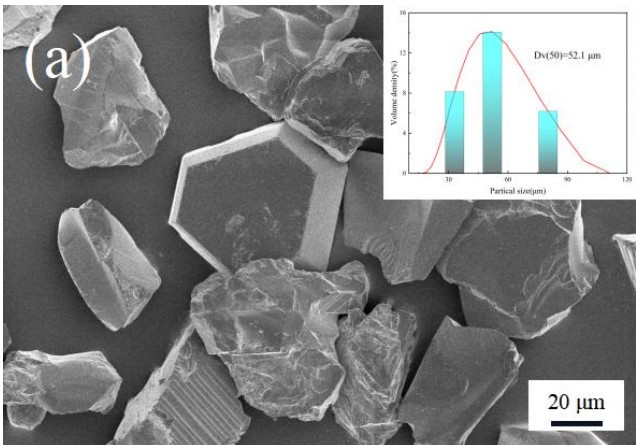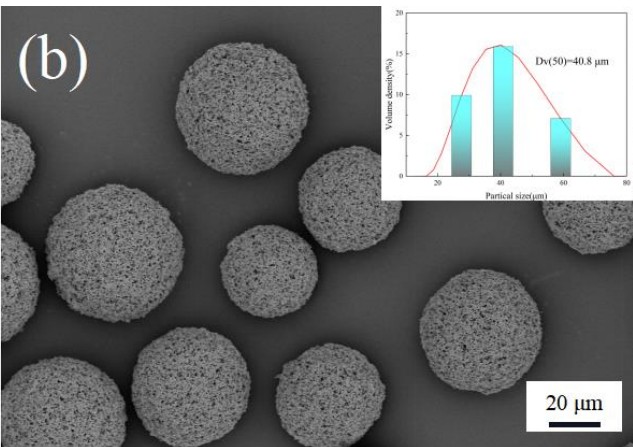

**Figure 2.** SEM and Laser particle-size analysis of (**a**) conventional micron powder and (**b**) granulated nano-alumina powder generated by atomization.

X-ray diffraction (XRD) (D8 Advance, Bruker, Germany) analysis of the nano- and micron-sized $Al_2O_3$ powders was carried out using Cu Ka radiation at a scanning speed of 6° per minute between $2\theta$ = 20 and 90° (shown in Figure 3). It can be seen from the figure that an $\alpha$-$Al_2O_3$ phase structure was obtained.

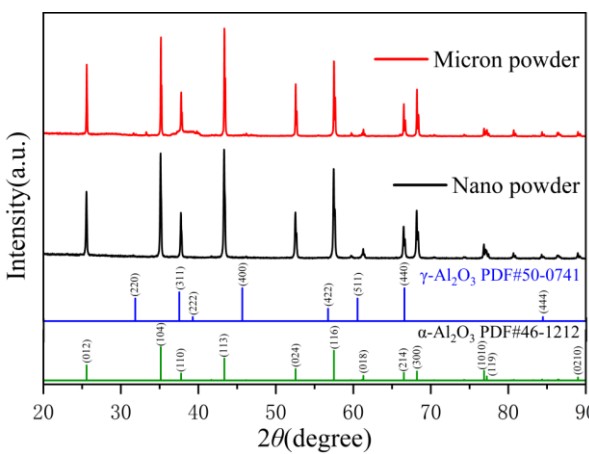

**Figure 3.** XRD pattern of $Al_2O_3$ powder.

### 2.2. Coating Preparation and Characterization

The substrate material was a square 2A12 aluminum alloy with a length of 40 mm and a height of 5 mm, and its elemental composition is shown in Table 1. Before spraying,

the substrate was ultrasonically cleaned with acetone to remove surface oil and dust and sandblasted with quartz sand to coarsen the surface, thus ensuring mechanical anchoring between the coating and the substrate.

**Table 1.** Chemical compositions of the 2A12 aluminum.

| Composition | Si | Cu | Mg | Zn | Mn | Ti | Ni | Fe | Al |
|---|---|---|---|---|---|---|---|---|---|
| Percentage, $w/w$ | ≤0.5 | 3.8~4.9 | 1.2~1.8 | ≤0.3 | 0.3~0.9 | ≤0.15 | ≤0.1 | ≤0.5 | Bal |

Before depositing the $Al_2O_3$ coating, a NiCrCoAlY transition layer with a thickness of about 100 μm was sprayed on the aluminum alloy substrate using German GTV plasma-spraying equipment to reduce the residual stress between the substrate and the $Al_2O_3$ ceramic due to the difference in thermal expansion coefficients. The plasma-spraying parameters are shown in Table 2.

**Table 2.** Process parameters of plasma spraying.

| Parameters | NiCrCoAlY | $Al_2O_3$ | Unit |
|---|---|---|---|
| Plasma gas (Primary-Argon) | 50 | 41 | NLPM |
| Carrier gas (Secondary-Hydrogen) | 9 | 14 | NLPM |
| Arc current | 530 | 580 | A |
| Arc voltage | 65 | 71 | V |
| Spray distance | 130 | 120 | mm |
| Powder feed rate | 30 | 20 | g/min |

Field emission scanning electron microscopy (SEM) (Supra55, Zeisi, Germany) was used to investigate the microstructural characteristics of the coatings. MicroEYE 3000 software was used to analyze porosity of coating cross-section images. The nano- and micro-sized coatings were characterized by X-ray diffraction analysis (D8 Advance, Broker, Germany) using Cu Ka radiation at a rate scanning speed of 6° per minute between $2\theta = 20$ and 90°. The three-dimensional morphologies of the coatings were obtained by three-dimensional shape profiler (GT-X, Broker, Germany), which was also used to measure the root-mean-square height of the surface (Rq). The hardness of the coatings was measured 20 times under 100 g (0.9807 N) for 15 s using a Vickers microhardness tester (MF 1000, Mega, China).

The thermal control properties (absorptance and emissivity) of the coatings were investigated by an ultraviolet-visible-near infrared spectrophotometer instrument (UV-3600, Shimadzu, Japan) and Dual-band emissivity measurement instrument (IR2, Wanyi, China).

### 3. Results and Discussion

*3.1. The Microstructure and Phase of the Coatings*

Figure 4 shows the XRD patterns of the NC and CC. The metastable $\gamma$-$Al_2O_3$ phase is the dominant phase of the alumina coatings. This is due to the molten $Al_2O_3$ droplets dropping onto the cold substrate and cooling rapidly (about $10^6$ K/s [31]) during the plasma-spraying process. Compared with $\alpha$-$Al_2O_3$, metastable $\gamma$-$Al_2O_3$ with lower critical free energy is more likely to nucleate during rapid solidification. In addition, to calculate the content of the $\alpha$-$Al_2O_3$ phase in the coatings, a semi-quantitative analysis was carried out by following Equation (1) for NC and CC [32].

$$C_\alpha = \frac{I(104) + I(113) + I(116)}{I(104) + I(113) + I(116) + I(311) + I(400) + I(440)} \tag{1}$$

where $C_\alpha$ represents the content of the $\alpha$-$Al_2O_3$ phase and $I$ denotes the area of the peak.

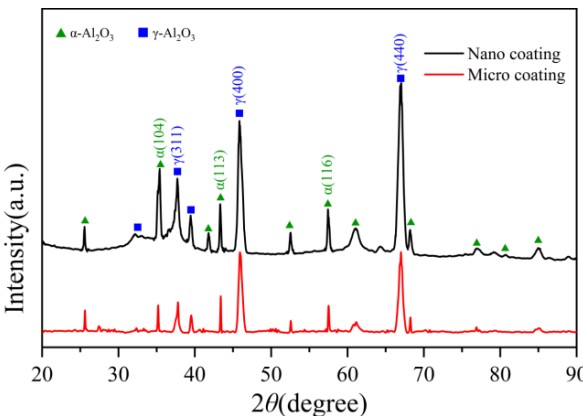

**Figure 4.** XRD pattern of Al$_2$O$_3$ coating.

The results show that the α-Al$_2$O$_3$ phase in NC (21.4%) is much higher than that in CC (12.7%). It was reported that the α-Al$_2$O$_3$ phase in these coatings is mainly derived from unmelted particles during spraying [33]. Compared with the micron-sized alumina powder, the granulated nano-alumina powder produced via atomization has a lower coefficient of thermal conductivity [24]. Therefore, the coating formed by the nano-alumina powder contains more unfused particles compared to the micron powder formed under the same spraying parameters, resulting in increased α-Al$_2$O$_3$ phase content.

Figure 5a,c show the surface morphology of CC and NC, respectively. From these figures, it is evident that the coatings exhibit lamellae structures of different shapes and sizes. In the spraying process, the molten droplets contact the substrate and spread out, leading to the formation of a lamellar morphology. As shown in Figure 5a, the lamellar area of the CC is larger compared to the NC, which is due to the different melting degrees of the powder [34]. The nanostructured coatings exhibit a two-state distributed microstructure, which consists of both fully melted lamellae and the partially melted structure of particles. This is similar to the coatings reported in the literature [34]. Further, the CC's surface has more cracks due to tensile stress exceeding the material's fracture strength during solidification [35], as shown in Figure 5c. However, in the nanostructured alumina coatings, partially melted zones can act as barriers for crack propagation but also cause greater porosity, as shown in Figure 5d. This effect has been extensively reported in the APS literature [36].

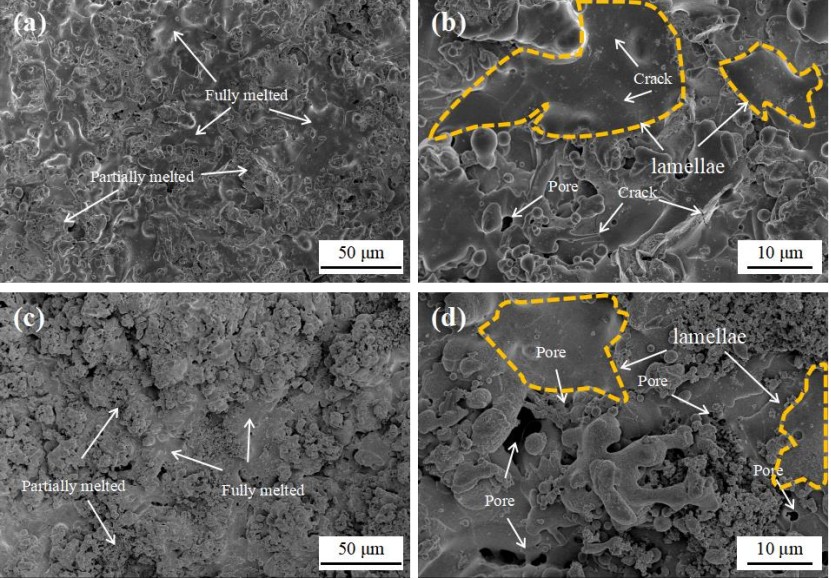

**Figure 5.** SEM images of (**a**,**b**) micro Al$_2$O$_3$ coating (CC) and (**c**,**d**) nano Al$_2$O$_3$ coating (NC).

Figure 6a,b depict the surface three-dimensional morphology of the CC and NC, respectively. The root-mean-square heights (Rq) of the surfaces of the CC and NC are 9.984 μm and 18.347 μm. According to the literature, the surface roughness of a coating is inversely proportional to the melting degree of the powder [37]. As a result, the nanostructured coating exhibits a higher degree of surface roughness due to the lower melting degree of the nanopowder.

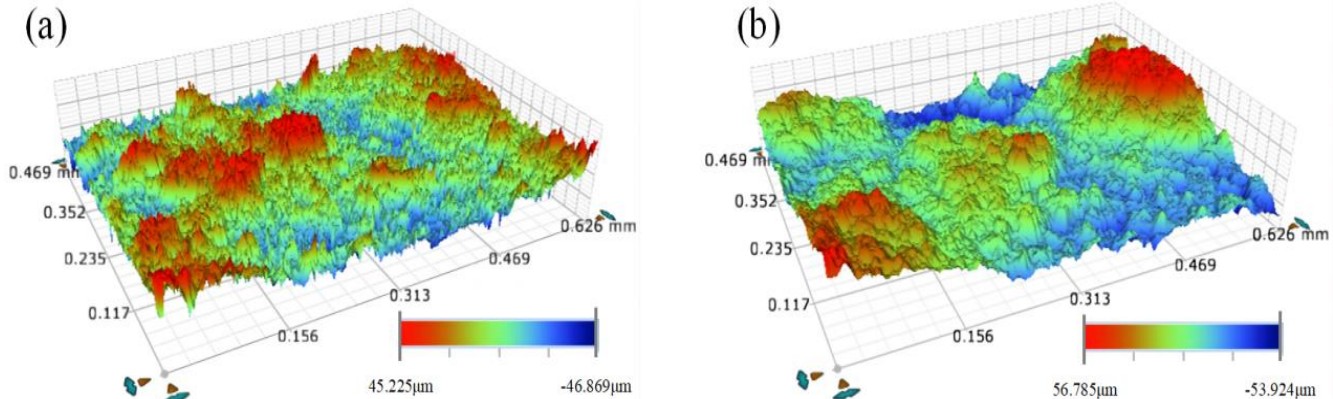

**Figure 6.** Surface morphology of (**a**) micro $Al_2O_3$ coating; (**b**) nano $Al_2O_3$ coating.

The cross-sectional morphologies of the coatings are shown in Figure 7. The interfaces among the substrate, the NiCrCoAlY bonding layer, and the ceramic coating can be observed. The thickness of the bonding layer in both coatings is 70–120 μm, the thickness of the ceramic coating of the CC ranges from 287–342 μm (Figure 7a), and the thickness of the ceramic coating of the NC ranges from 285–317 μm (Figure 7c). The porosity of the coating was measured by MicroEYE 3000 software. The average porosity observed in the CC was 3.48% compared to 5.21% for the NC.

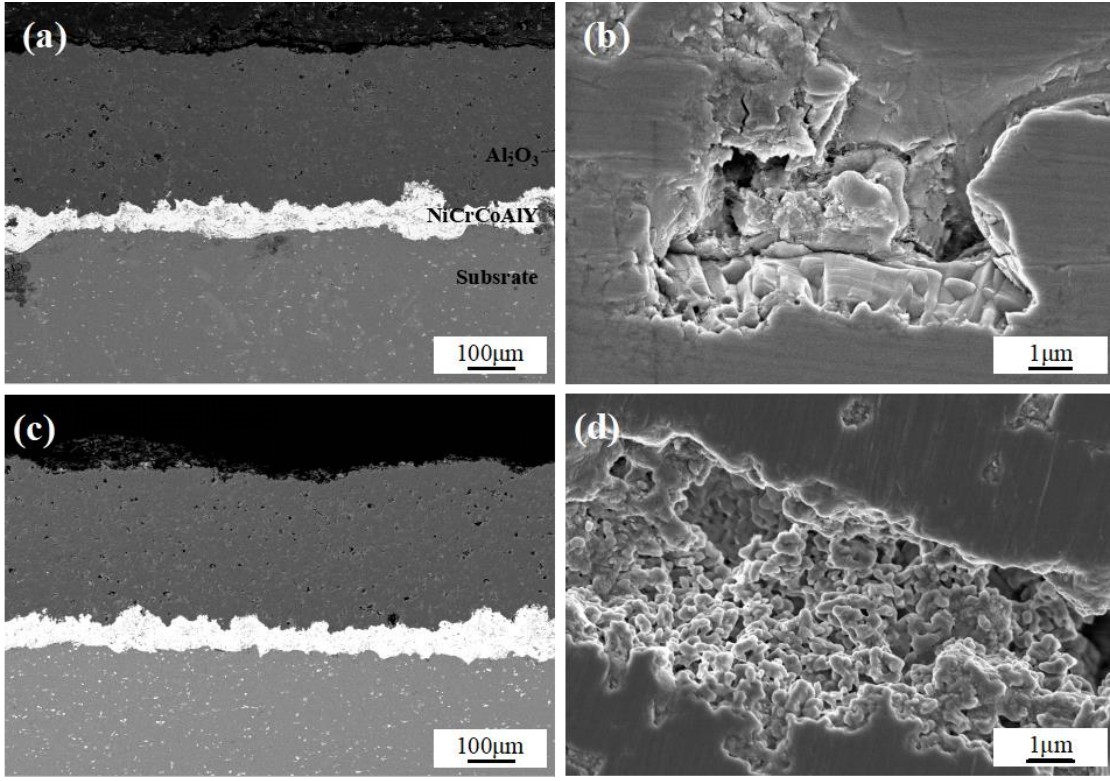

**Figure 7.** Cross-sectional morphology of (**a**,**b**) micro $Al_2O_3$ coating and (**c**,**d**) nano $Al_2O_3$ coating.

Figure 7b,d show high-magnification FESEM images of the pores of the plasma-sprayed CC and NC, respectively. It can be observed that the pores of the CC show the typical lamellar structure of a plasma-sprayed coating (Figure 7b), whereas partially/unfused nanoparticles can be seen in the pores of the NC with the fully melted lamellae bonded together in a mesh (Figure 7d). The microhardness of the coatings was measured using a Vicker's micro hardness tester, with a load of 0.9807 N applied to the polished cross-sections of the coatings for 15 s. The average microhardness values for the CC and NC were 1079.7 HV and 1168.8 HV, respectively. The average micro hardness of the NC is higher than the CC due to the following reasons: (1) At low loads, the microhardness of the coatings depends largely on their phase composition rather than coarse pores, cracks, etc. [38]. (2) According to the Hall-Petch theory, the increase in the hardness of an NC is also related to its nanograin size [39].

Weibull statistical analysis was used to further analyze the hardness parameter, as shown Figure 8 [40]. Figure 8a shows a single linear distribution of the fitted Weibull curve for the CC, which is due to the essentially uniform coating organization and the small variation in the hardness values. However, as shown Figure 8b, the Weibull curve fitted for the NC consists of two lines with different slopes: section A—low hardness due to poor interparticle bonding (cohesion); section B—high hardness due to the presence of unfused particles with an $\alpha$-$Al_2O_3$ phase. The difference in the hardness distribution of the coatings is also verified by the microstructure; the hardness distribution of the conventional coating reflects its micromorphology, which is basically composed of completely melted laminates, while the hardness distribution of the nano-coating is related to the two-state distribution of the microstructure.

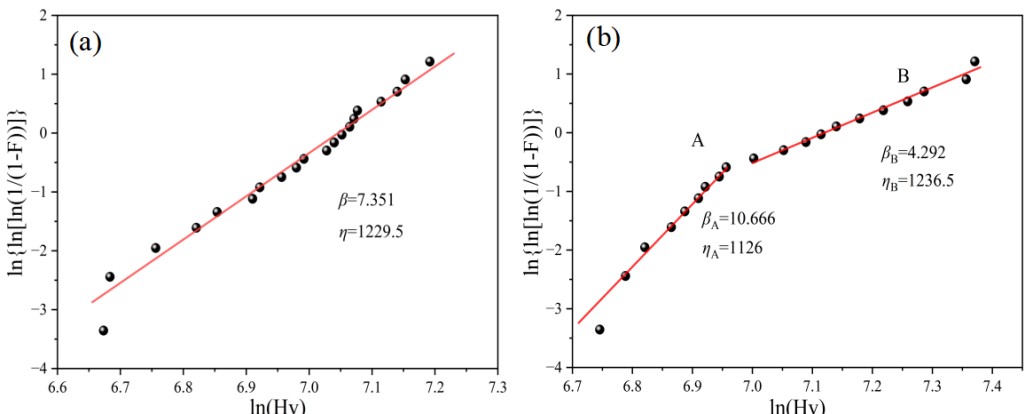

**Figure 8.** Weibull distribution of the hardness test values of (**a**) micron $Al_2O_3$ coating (**b**) nano $Al_2O_3$ coating.

### 3.2. Thermal Control Property

Figure 9 shows the reflection spectra of the alumina coatings (NC and CC). The CC has a reflectance coefficient of 70–75% within 417–1418 nm (Figure 9a). Its main absorption boundary is located at 285 nm (Figure 9b). The reflectivity of the NC within 446–1586 nm is more than 75% (Figure 9a). Its maximum absorption peak is located at 227 nm (Figure 9b). The micron-alumina coating has a reflectance coefficient of 70–75% within 417–1418 nm. Over the entire wavelength range, the solar absorptance ($\alpha_s$) values of the nanostructured and conventional alumina coatings are 0.26 and 0.324. In the NIR band, this difference can be caused by adsorption via gas. However, in the short-wavelength region (in the range of 200–400 nm), the absorption difference is caused by the band gap of the material [9]. According to Planck's law, i.e., $\lambda \geq hc/E_g$, ($h$ is Planck's constant), for intrinsic absorption to occur, the photonic energy must be equal to or greater than the band gab ($E_g$).

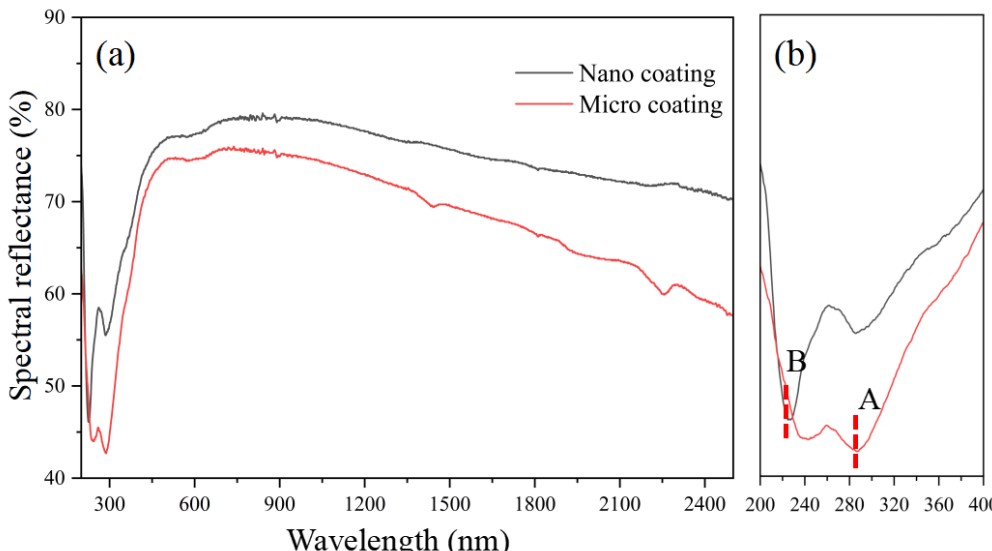

**Figure 9.** Reflectivity spectra of $Al_2O_3$ coatings in the wavelength range of (**a**) 200–2500 nm and (**b**) 200–400 nm.

As shown in Figure 10, the band gap corresponding to the maximum absorption peak of the alumina coating was calculated by the Kubelka-Munk method. The band gap of the CC near 285 nm is 3.71 eV (Figure 10A) and the maximum absorption peak of the NC corresponds to a band gap of 5.12 eV (Figure 10B). The blue-shift phenomenon of the main absorption peak in the NC is caused by the increase in the $\alpha$-$Al_2O_3$ phase. Therefore, the reflectance of the NC is significantly increased compared to the CC within 227–285 nm. This result is consistent with previous reports in the literature [30].

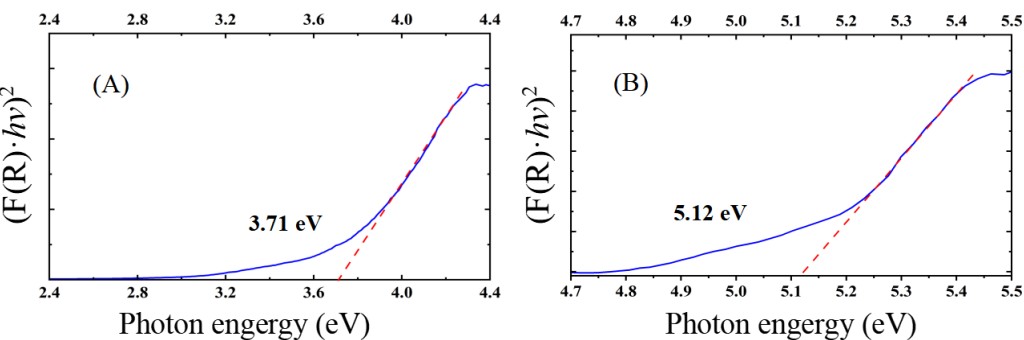

**Figure 10.** Determination of the maximum absorption peak of (**A**) CC and (**B**) NC corresponding to band gap determined via Kubelka-Munk method.

In addition, differences in the microscopic morphology of the coating can also cause differences in spectral reflectance. When solar radiation is incident on the surface of an alumina coating, unlike metal coatings, wherein reflection, absorption, and emission occur only at the thin interface, alumina coatings have low absorption and high transmission at the interface [41,42]. Therefore, the reflectivity of alumina coatings depends on the volume of absorption and the scattering properties of the radiation.

As shown in Figure 11, plasma-sprayed nano-alumina coatings typically have a heterogeneous, usually porous, microstructure [29]. When radiation enters the interior of the coating, volume scattering often occurs due to the optical index gaps between the air in the pores and the matrix, which results in the modification of the direction of radiation propagation around the point of heterogeneity and multiple reflections at the interface between heterogeneity and the matrix medium. Some radiation scatters and finally returns to the surface, which can contribute to the reflectivity of the coating. In fact, the intensity and direction of scattered radiation is determined by the inhomogeneity of the medium's

density and the internal particle scale. The internal scattering of a coating is enhanced with the increase in porosity. In addition, the nanostructured coating has a large number of nanoparticles, as shown in Figure 7d, and backscattering occurs when radiation encounters these particles, which also improves the reflection performance of the coating. Therefore, the NC shows high reflective performance and its degree of volume scattering is enhanced due to the presence of numerous pores and nanoparticles inside.

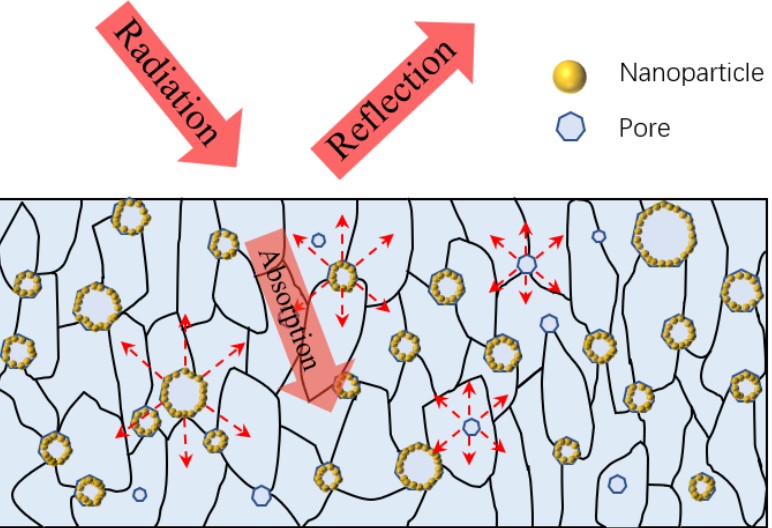

**Figure 11.** Schematic diagram of radiation in nano-alumina coating.

Figure 12 shows the temperature dependence of the emissivity ($\varepsilon$) of the CC and NC in 1–22 μm wavebands. From the figure, the NC and CC have higher emissivity values than some forms of PEO and anodizing coatings applied to aluminum alloys [14,43], ranging from 0.863 to 0.870 and 0.875 to 0.883, respectively, at different temperatures. According to Kirchhoff's law, for an opaque object under thermal equilibrium conditions: $\varepsilon = \alpha = 1 - \rho$ ($\alpha$ and $\rho$ are the absorption rate and reflectance of the sample with respect to the projected energy, respectively) [44]. Therefore, the emissivity of the NC is reduced due to its high-reflectance property. In addition, due to the excellent thermal stability of alumina, the plasma-sprayed alumina coating's emissivity is less affected by thermal effects, and the variation is less than 0.01 in the range of 300–393 K.

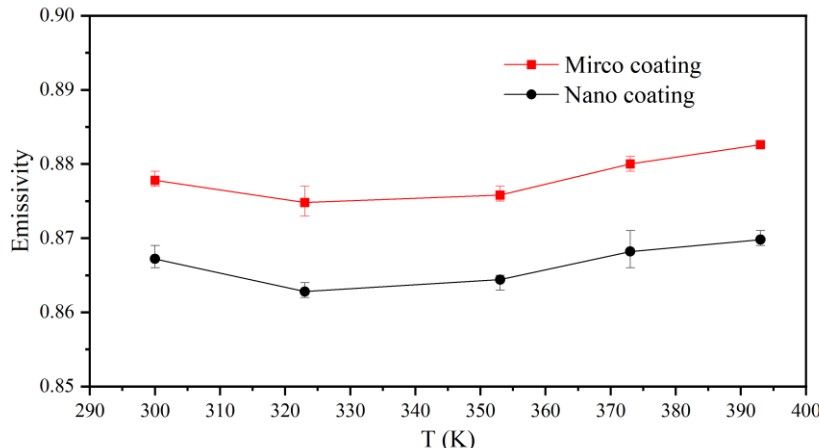

**Figure 12.** The temperature dependence of the emissivity ($\varepsilon$) of $Al_2O_3$ coatings in 1–22 μm wavebands.

## 4. Conclusions

In this work, nanostructured and conventional alumina coatings were successfully deposited on an aluminum alloy substrate (2A12) using the plasma-spraying technique. The

hardness and thermal control properties of the coatings prepared from different alumina powders were compared and analyzed. Compared to the CC, the hardness of the NC increased from 1079.7 Hv to 1168.8 Hv, solar absorptance decreased from 0.324 to 0.26 within 200–2500 nm, and the emissivity difference was about 0.012 within 300–393 K.

These results are related to the two-state distribution in the nano-alumina coating, which can be explained as follows. (1) The microstructure of the two-state distribution causes the coating to produce more $\alpha$-$Al_2O_3$ phases, while the increase in the content of $\alpha$-$Al_2O_3$ causes the absorption boundary to shift to a short wavelength direction such that the absorption of radiation by the coating is reduced. (2) The bimodal microstructure leads to the existence of a large number of pores in the coating, and the nanoparticles promote the volume reflection of radiation, which improves the reflective performance of the coating. The results of this study demonstrate the potential of the plasma-spraying process towards fabrication of thermal control coatings.

**Author Contributions:** Investigation, G.H.; resources, J.Z.; data curation, G.H.; writing—original draft preparation, G.H.; writing—review and editing, W.G., L.J. and Y.H.; supervision, D.H., Z.L. and L.J.; funding acquisition, Z.X. All authors have read and agreed to the published version of the manuscript.

**Funding:** This research was funded by the General program of the National Natural Science Foundation of China] grant number [52130509] And The APC was funded by [Lei Jia].

**Conflicts of Interest:** The authors declare that they have no known competing financial interests or personal relationships that could have appeared to influence the work reported in this paper.

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
