# Peer review of "Study of the Mechanical Properties and Thermal Control Performance of Plasma-Sprayed Alumina Coating on Aluminum Alloy Surface"

_applsci, doi:10.3390/app13020956_

Round 1

Reviewer 1 Report

The authors can address some of the queries listed in the attached pdf.

Author Response

请参阅附件

Reviewer 2 Report

The presented paper describes a solid work devoted to the important and novel problem of thermal control coatings improvement to reduce the absorption of solar radiation, increase the surface thermal radiation, and reduce the thermal equilibrium temperature of the spacecraft.

 In present work authors used atmospheric plasma spraying method to prepare conventional alumina coatings and the nanostructured alumina coatings, respectively, to characterize the microstructure, thermal control properties and mechanical properties of different alumina coatings, and to discuss the relationship between microstructure and composition with thermal control properties and mechanical properties.

Generally the authors achieved the stated goals and improved the parameters of the alumina coatings.

On my opinion the present work is useful for the community.

 The English language of the paper is generally well, I suggest minor spell check.

 I recommend that this paper is accepted after minor spell check.

Author Response

  1. 对评论的回应:(建议修改 1:建议进行轻微的拼写检查。

1).回应:回应:作为审稿人建议,我们已经纠正了稿件中的拼写错误,并再次检查了稿件。修改如下:

“X射线衍射(XRD)(D8 Advance,Broker,德国)使用Cu Ka辐射以每分钟6°的速率扫描速度在 = 20°和90°之间分析纳米和微米尺寸的Al2O 3粉末如图3所示”修改为“X射线衍射(XRD)(D8 Advance,布鲁克,德国)分析使用Cu Ka辐射以2θ = 20 和 90° 之间的每分钟 6° 如图 3 所示。

“图2.扫描电镜和激光粒度分析仪将(a)常规微米粉体和(b)粒状纳米氧化铝粉经雾化“修改为”图2。通过雾化对(a)常规微米粉末和(b)颗粒状纳米氧化铝粉末进行SEM和激光粒度分析”。

“半稳态γ-Al2O3相是氧化铝涂层的主相”修改为“亚稳γ-Al2O3相是氧化铝涂层的主相”。

“在NIR波段中,这种差异可能是由化学气体吸附引起的”修改为“在NIR波段中,这种差异可能是由化学气体吸附引起的。

“根据普朗克定律,λhc/E g,(h是普朗克常数),要发生本征吸收,光子能量必须等于或大于带状带(E g)”修改为“根据普朗克定律,λhc/E g,(h是普朗克常数),要发生本征吸收,光子能量必须等于或大于带状嘎b(E g)”。

“NC中主吸收峰的蓝移现象,由α-Al 2 O 3相的增加引起”修改为“NC中主吸收峰的蓝移现象,是由α-Al2O3相的增加引起的。

“仅在薄界面发生吸收和发射”修改为“仅在薄界面发生吸收和发射”

谢谢你,最诚挚的问候。

你的真诚,何博士

Reviewer 3 Report

- In this work, two kinds of Al2O3 powders were used for coating: regular micron powder and nanopowder after granulation. The article does not provide data on the preparation of the granulated powder composition.

- line 141-143: «In addition, to calculate the content of α-Al2O3 phase in the coatings, semi-quantitative analysis was carried out by following equations (1), for NC and CC [32]». How reliable are these calculations? It is necessary to refer to sources that justify the correctness and applicability of this formula for quantitative phase analysis.

- line 274-278 «This is related to the increase of α-Al2O3 phase content and the porous and duplex structure of the coating. With the increase of α-Al2O3 phase content, the absorption boundary moves to the short-wavelength direction, and the radiation absorption of the coating decrease. The increased porosity and two-state distributed microstructure contribute to the volume scattering of radiation, which also improves the reflection performance of the coating.». The paper does not present the results of microstructure studies showing the duplex structure or the two-part distributed microstructure of α-Al2O3 coatings.

- Did the study of hardness and thermal management properties of the coatings take into account the effects of coating roughness?
